:ΘΣ PLOS | ONE

# Changes in airway diameter and mucus plugs in patients with asthma exacerbation

Yuki Yoshida[1,2], Yotaro Takaku[1]*, Yasuo Nakamoto[1], Noboru Takayanagi[1], Tsutomu Yanagisawa[1], Hajime Takizawa[2], Kazuyoshi Kurashima[1]

1 Department of Respiratory Medicine, Saitama Cardiovascular and Respiratory Center, Kumagaya City, Saitama, Japan, 2 Department of Respiratory Medicine, Kyorin University School of Medicine, Mitaka city, Tokyo, Japan

☉ These authors contributed equally to this work.
* takaku.yotaro@pref.saitama.lg.jp

## Abstract

### Background

Airway obstruction due to decreased airway diameter and increased incidence of mucus plugs has not been directly observed in asthma exacerbation. We studied the changes in the inner diameter of the airway (Din) and the frequency of mucus plugs by airway generation in patients with asthma exacerbation. We compared these patients to those in a stable phase using high-resolution computed tomography (HRCT).

### Methods and findings

Thirteen patients with asthma were studied by HRCT during asthma exacerbation and in a stable period. The HRCT study was performed on patients who could safely hold their breath for a short while in a supine position 1 hour after initial treatment for asthma exacerbation. Using a curved multiplanar reconstruction (MPR) software, we reconstructed the longitudinal airway images and the images exactly perpendicular to the airway axis to measure the Din and mucus plugs from the second- (segmental) to sixth-generation bronchi in all segments of the lungs.The ratios of Din (exacerbation/stable) were 0.91($P = 0.016$), 0.88 ($P = 0.002$), 0.83 ($P = 0.001$), 0.80 ($P = 0.001$), and 0.87 (NS) in the second-, third-, fourth-, fifth-, and sixth-generation bronchi, respectively. The percentages of airway obstruction due to mucus plugs were notably higher in the fourth- and fifth-generation bronchi (17.9%/18.1% in stable phase and 43.2%/45.9% in the exacerbation phase, respectively) than in the other generations of bronchi.

### Conclusions

Among the bronchi examined, the fourth- and fifth-generation bronchi were significantly obstructed during asthma exacerbation compared with the stable phase in terms of a decreased airway diameter and mucus plugs.

**Data Availability Statement:** All relevant data are within the paper and its Supporting Information files.

**Funding:** The authors received no specific funding for this work.

**Competing interests:** The authors have declared that no competing interests exist.

## Introduction

In asthma exacerbation, the Global Initiative for Asthma (GINA) guideline recommends inhalation treatments with a short-acting beta-2 agonist (SABA) and ipratropium bromide in addition to systemic corticosteroids [1]. However, most inhaled therapies do not reach the small airways (airways with internal diameter < 2 mm) that comprise multiple aspects of asthma [2–8]. Also, mucus plugs are considered to be one of the mechanisms of airway obstruction in fatal asthma [9,10]. If the inner diameter of the airway (Din) and mucus plugs could be directly observed in acute asthma, it would help us understand how inhalers act on airways in asthmatics. The mucus plugs, as well as narrowed airways, could be the target of inhaled corticosteroids and bronchodilators during asthma exacerbation.

Recently, a quantitative image analysis of airway obstruction due to mucus plugs has been explored in severe asthma [11]. Quantitative computed tomography (CT) imaging can provide structural and functional information on asthma [12–16]. The recently developed CT technique known as curved multiplanar reconstruction (MPR) can visualize longitudinal airway images and accurately analyze short-axis images of small airways that cannot be recognized in standard high-resolution CT (HRCT) images. Using such a technique, we have shown that small airways with a 1.5-mm Din on a conventional multidetector HRCT or a 0.8-mm Din on ultra-HRCT can be measured in clinical situations [17, 18]. Airway analysis of MPR images can be used to measure mucus plugs quantitatively as well.

The purpose of this study is to evaluate the changes in Din and the frequency of airway occlusion by mucus plugs from the segmental bronchus (second generation) to the sixth-generation bronchi by comparing HRCT data during asthma exacerbation with those obtained in the stable phase. The findings might provide a new dimension in quantifying asthma severity and asthma heterogeneity.

## Materials and methods

### Study design and participants

The Institutional Review Board of Saitama Cardiovascular and Respiratory Center approved this study (IRB No. 2012 033), and written informed consent was obtained from all study patients. CT scans of the subjects were obtained during asthma exacerbation and in the stable phase during the follow-up period. There was over 6 months between the CT scans in the exacerbation phase and the recovery phase. The asthma severity in the stable phase and exacerbation severity were classified according to the GINA guidelines [1]. We evaluated asthma control according to the consensus-based GINA symptom control tool. All patients during the asthma exacerbation phase visited our clinic due to the worsening of wheezing with dyspnea and were diagnosed with asthma exacerbation. The severity of asthma exacerbation in this study's patients was determined according to the descriptions in the section "Management of asthma exacerbations in the emergency department" in the GINA guidelines. We used objective assessment and other measurements, such as respiratory rate, pulse rate, $O_2$ saturation, and accessory muscles being used, for the determination. The patients with asthma exacerbation with life-threatening signs, as assessed by the doctor in charge, were excluded from this study. The CT study was performed 1 hour after initial treatments to evaluate the airway obstruction and screen for other comorbidities that mimic asthma exacerbation, such as bronchopneumonia or heart failure. In the stable phase of asthma, a CT study was performed to assess airway remodeling and airway dimensions. Stable asthma was defined as the absence of clinic visits for asthma exacerbation, unchanged use of asthma medication for maintenance therapy, and the stable use of rescue medication (no more than four puffs per day of a short-acting bronchodilator) during the previous month.

## CT data acquisition and image analyses

A 256-slice CT scanner (Brilliance iCT; Philips Healthcare, Cleveland, OH, USA) was used as previously described [17]. The CT scans were obtained from the suspended end-inspiratory volume at baseline and during exacerbation. We also checked lung size and lung architecture in the same slice to confirm they were obtained at the end-inspiratory volume. All CT row data sets were analyzed using an MPR software program (SYNAPSE®3D, Fujifilm Co. Tokyo, Japan), where the Din and mucus plugs were measured per airway generation for all 18 segments of the lung. We first selected a target airway and a point in the airway lumen at the level of the segmental bronchus, that is, the second-generation bronchus according to the Japanese Cancer Society [19]. From this point, the airway tree was identified using a trained detector and a spanning tree algorithm, and we obtained the selected bronchial pathway to the point of the sixth-generation bronchus. If a mucus plug obstructed the bronchus before the sixth-generation bronchus, we could trace the bronchus manually to the sixth-generation bronchus. The bronchus was selected on the principle that it should be more centered and well recognized in one segment. The same bronchial trees were used for analysis during the stable phase and at exacerbation. We were then able to obtain short-axis images that were exactly perpendicular to the long axis of the airway at each generation bronchus. From the centroid point of the lumen, rays fanning out over 360 degrees were examined to determine the inner airway walls along the rays using the full width with the half-maximum principle. We were then able to obtain the mean values for the Din. Mucus plug (mucus occlusion) was defined as the complete occlusion of a bronchus, contiguous with patent airway lumen on the longitudinal airway image (Fig 1). Mucus plugs were distinguished from focal opacified airways, such as airway collapse or adjacent blood vessels, identifying no focal decrease of outer airway caliber and recognizing adjacent blood vessels between opened proximal and distal airways from the mucous area. We checked whether mucus plugs were associated with bronchiectasis, defined as a

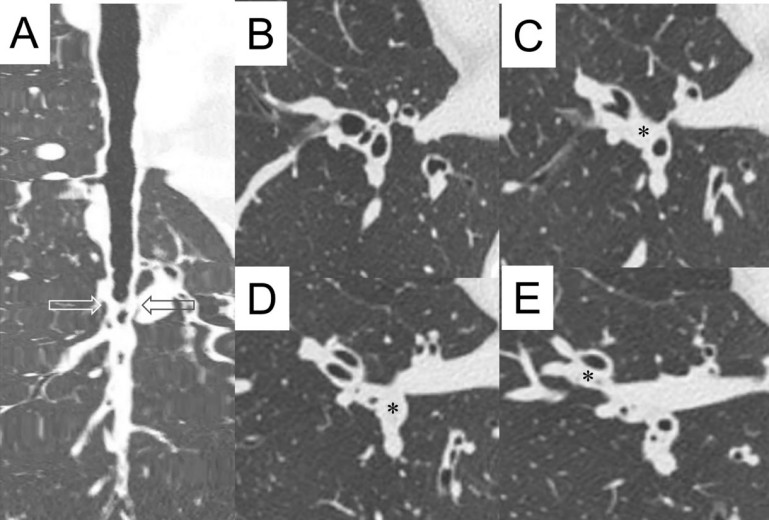

**Fig 1. Representative view of mucus plugs.** A: Longitudinal view of right B10 bronchus. Peripheral parts of the bronchus (bottom of the panel) are open. B: Conventional HRCT images of right B7, B8, B9, and B10. C: Conventional HRCT image just below the level of panel B. Occluded segmental bronchus (second generation, marked with an asterisk) of right B9 is shown. D: Conventional HRCT image just below the level of panel C (indicated level in panel A by arrows). Occluded segmental bronchus (second generation, marked with an asterisk) of right B10 is shown. E: Conventional HRCT image just below the level of panel D. Occluded subsegmental bronchus (third generation, marked with an asterisk) of right B8 is shown.

bronchoarterial ratio of greater than 1.5. We noted the presence of mucus plugs as "yes" or "no" from the second- to sixth-generation bronchial compartments in one selected airway in one segment. We could then calculate the percentage of mucus plug in a given bronchial generation in a given lung segment. The validation of these CT measurements has been described by [17].

## Pulmonary function tests

Pulmonary function tests (PFTs) were performed within two weeks of obtaining the HRCT scans in the stable phase. PFTs were performed according to the guidelines of the American Thoracic Society [20]. Spirometry parameters (forced expiratory volume in 1 second (FEV1) and FEV1/forced vital capacity (FVC)) were measured with a CHESTAC8800 (Chest Inc., Tokyo, Japan). Spirometry was done before and 20–30 minutes after the inhalation of 200 g of salbutamol, and the values after the bronchodilator were used for baseline data.

## Statistical analyses

Mean Din and the number of mucus-plug-positive bronchial compartments were calculated in the second- to sixth-generation bronchial compartments in all 18 segments of the lung in 13 subjects. These values were compared by total, by bronchial generations, or by lobes between the stable phase and acute phase. The mean Din of the total compartments and the total number of mucus-positive bronchial compartments in the stable phase were analyzed for correlation to spirometry valuables in the stable phase. Data are expressed as the mean ± standard deviation (SD) or as the median (range), as appropriate. A paired t test was used to compare the Din in the stable and acute phase. Statistical analyses between groups were first performed with the Kruskal–Wallis test followed by Dunn's test for comparisons between groups. A value of $P < 0.05$ was considered to be significant. The Prism 5 software program (GraphPad Software, Inc., La Jolla, CA, USA) was used for the analyses.

## Results

Seventeen patients who had emergency visits to our center with asthma attacks were included. Of these, three were excluded because of their life-threatening condition. One did not consent, and the remaining 13 patients were enrolled. The patient characteristics are reported in Table 1. There was no notable difference in the number of men and women in the subjects. Events of asthma exacerbation were classified as mild/moderate and severe, and four patients were hospitalized for further treatment. When CT studies were performed after initial treatments, no patients showed any life-threatening signs or orthopnea. All patients could hold full inspiration for a while. The patients' asthma was well (61.5%) or partly controlled (38.5%) in the stable phase.

In 13 patients, five bronchial compartments from segmental (second generation) to the sixth-generation bronchus of the representative airway for one segment were analyzed in 18 segments of the lung in the stable phase and during an asthma exacerbation. As shown in Fig 1, Din was not measured in mucus-occluded parts.

Fig 2 shows a representative view of the bronchial trees of the same patient in the stable phase and during exacerbation. The tracheal smooth muscles constricted, and the tracheal cartilage is visible from the upper to the middle part of the trachea during asthma exacerbation. Parts of the peripheral airways could not be traced, and middle-zone airways were thin or occluded during exacerbation.

Fig 3 shows ratios of Din during exacerbation and the stable phase. Each point represents the mean Din ratio of 13 subjects for each segment. The mean Din for all segments of 13

**Table 1. Patient characteristics.**

| Number of subjects | 13 |
|---|---|
| Age (years) | 56 (± 9.8) |
| Gender, F:M | 7:6 |
| Smoking history (number) | Never: 7 |
| | Ex: 6 |
| | Current: 0 |
| Asthma duration (years) | 3.5 (1.6–13.4) |
| Asthma medication | SABA: 3 |
| | ICS: 1 |
| | ICS/LABA: 3 |
| | ICS/LABA/LTRA: 1 |
| | ICS/LABA/LTRA/TP: 2 |
| | ICS/LABA/LTRA/TP/OCS: 2 |
| | ICS/LABA/LTRA/TP/OCS/anti-IgE: 1 |
| Asthma severity at stable (number) | Well-controlled: 8 |
| | Partly controlled: 5 |
| Exacerbation severity (number) | Mild or moderate: 9 |
| | Severe: 4 |
| | Life-threatening: 0 |
| Body mass index (kg/m$^2$) | 24.3 (±4.3) |
| Baseline FEV1(% predicted) | 82.2 (±16.4) |
| Baseline FVC (% predicted) | 87.1 (±17.4) |
| Baseline FEV1/FVC (% predicted) | 85.3 (±14.9) |
| Bronchodilator reversibility (%) | 5.4 (±4.7) |
| Baseline FeNO (ppb) | 47.2 (±32.6) |
| Baseline eosinophils in blood (×10$^3$ μ/L) | 0.54 (±0.60) |
| Exacerbation eosinophils in blood (×10$^3$ μ/L) | 0.83 (±0.99) |

FeNO, fractional exhaled nitric oxide; FEV1; forced expiratory volume 1 second; FVC, forced vital capacity; ICS, inhaled corticosteroid; LABA, long-acting beta-2 agonist; LTRA, leukotriene receptor antagonist; OCS, oral corticosteroid; SABA, short-acting beta-2 agonist; TP, theophylline

patients were 3.6 ± 0.5, 2.6 ± 0.3, 2.2 ± 0.3, 1.8 ± 0.2, and 1.8 ± 0.2 mm, respectively, in the second- to sixth-generation bronchi at asthma exacerbation. The ratios of the Din were notably reduced in the second-, third-, fourth-, fifth-, and sixth-generation bronchi (0.91 ± 0.08, $P = 0.016$; 0.88 ± 0.13, $P = 0.002$; 0.83 ± 0.11, $P = 0.001$; 0.80 ± 0.15, $P = 0.001$; 0.87 ± 0.13, NS, respectively).

The ratios of mucus-occluded bronchial compartments in 13 subjects by airway generation during asthma exacerbation and the stable phase are shown in Fig 4. Each point represents the ratio of mucus-occluded compartments in 13 patients for each segment.

In the stable phase of asthma, mucus occlusion was observed in 17.9% of the fourth-generation bronchi and 18.1% in fifth-generation bronchi. Mucus plugs are most notable in the fourth- and fifth-generation bronchi and in the lower lobes (left panels of Fig 4 and Fig 5). During the exacerbation phase of asthma, mucus plugs are observed in 43.2% of the fourth-generation bronchi and 45.9% in fifth-generation bronchi. Mucus plugs are most notable in the fourth- and fifth-generation bronchi and in the lower lobes (right panels of Fig 4 and Fig 5). Interestingly, the levels of airway where the mucus occlusion is most frequently observed are different among the lung lobes, especially during asthma exacerbation. In the upper lobes,

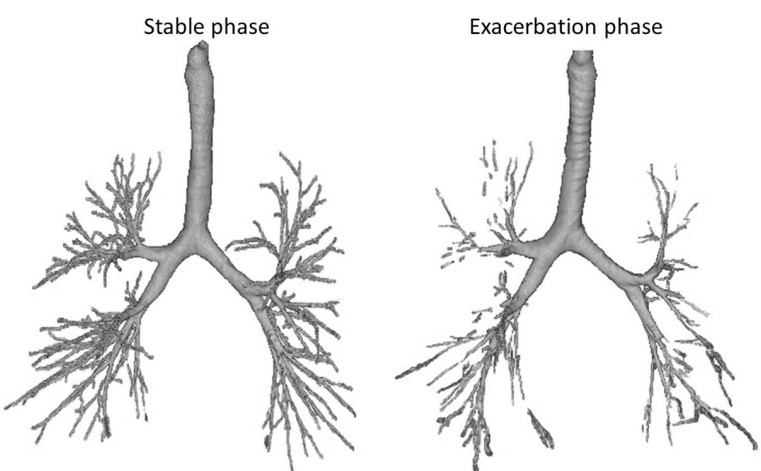

**Fig 2. Representative view of bronchial trees by HRCT of the same asthma patient in the stable phase and asthma exacerbation.** During the exacerbation phase, the tracheal smooth muscles are constricted, and the tracheal cartilage is visible. Parts of the peripheral airways could not be traced, and middle-zone airways are thin or occluded at exacerbation.

the frequency of mucus plugs is higher in the fifth- and sixth-generation bronchi, but in the lower lobes, the frequency of mucus plugs is higher in the fourth- and fifth-generation bronchi than in others (S1 Fig and S2 Fig). These mucus plugs occurred in the absence of bronchiectasis.

The relationship between CT image parameters and airflow obstruction by spirometry could be analyzed in the stable phase (Table 2). The mean Din for all compartments or by airway generation (data not shown) did not significantly correlate with the predicted percentage of FEV1 or FEV1/FVC, but the total number of mucus-occluded bronchial compartments for

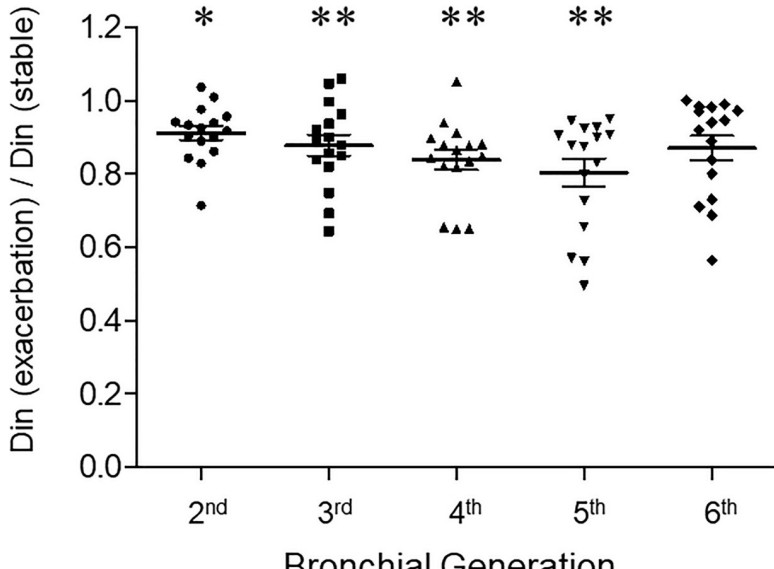

**Fig 3. The ratios of Din during asthma exacerbation and in the stable phase observed from the second- (segmental) to sixth-generation bronchi.** Din was measured per airway generation in all segments of the lung by HRCT for each subject at suspended end-inspiratory volume using curved MPR software. Each point represents the mean Din ratio of 13 subjects for each segment. * denotes $P < 0.05$ and ** denotes $P < 0.01$.

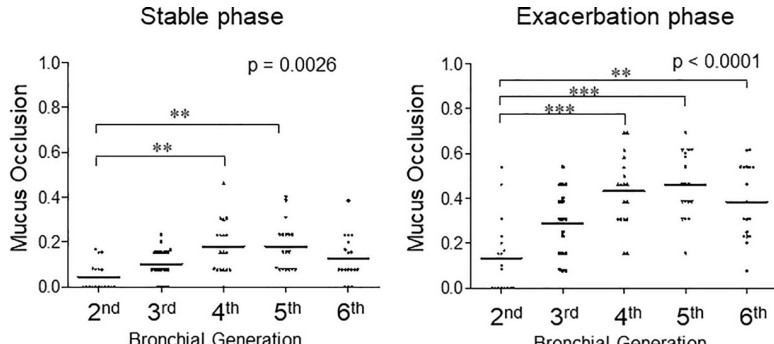

**Fig 4. The ratio of mucus-occluded bronchial compartments in 13 subjects by airway generation during asthma exacerbation and the stable phase.** Mucus occlusion (mucus plug) was measured by HRCT using curved MPR software. Mucus plugs are defined as the complete occlusion of a bronchus, contiguous with patent airway lumen on longitudinal airway image. Mucus occlusions are counted per airway generation in all segments of the lung for each subject. Each point represents the ratio of mucus-occluded compartments in 13 patients for each segment.

all compartments significantly correlated with the predicted percentage for FEV1 ($P = 0.0162$) and FEV1/FVC ($P = 0.0064$).

The linear regression model reports a coefficient (95% CI) for spirometry values. FEV1 and FVC denote forced expiratory volume for 1 second and forced vital capacity, respectively. The inner airway diameter (Din) and presence of mucus plugs are measured in the second- to sixth-generation bronchial compartments in the selected airway (see text) in each of the 18 segments of the lung. The total bronchial area Din represents the mean Din of the total bronchial compartments, and the total bronchial area mucus plug represents the total number of mucus-plug-positive compartments.

Regarding exacerbation severity, nine cases were classified as mild/moderate exacerbation, and four cases were classified as severe exacerbation. The mean Din for all compartments in mild/moderate exacerbation cases and severe cases were not different in the stable phase ($2.6 \pm 0.8$ mm versus $2.5 \pm 0.8$ mm, NS), but the mean Din was smaller in patients with severe exacerbation ($2.5 \pm 0.8$ mm$^2$ versus $2.3 \pm 0.7$ mm, $P = 0.042$). The average percentage of mucus plugs in the total measurement points in mild/moderate exacerbation cases and severe cases were $35.1\% \pm 19.6\%$ versus $29.8\% \pm 27.5\%$, respectively, during the exacerbation phase (NS).

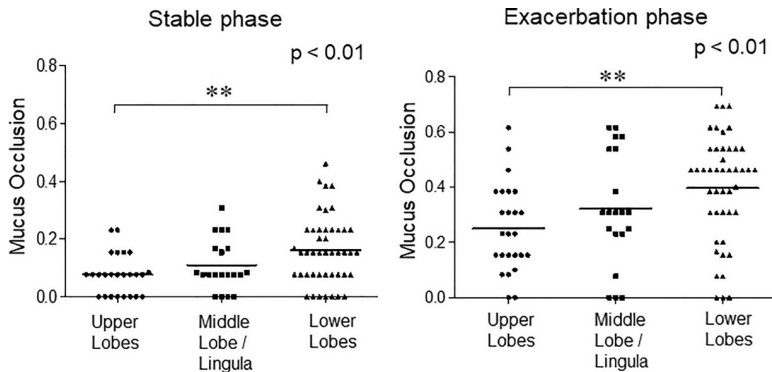

**Fig 5. The ratio of mucus-occluded bronchial compartments in 13 subjects during asthma exacerbation and the stable phase observed in the upper, middle/lingula, and lower lobes.** Mucus occlusion (mucus plug) was measured by HRCT using curved MPR software. Mucus occlusions were counted per airway generation in all segments of the lung for each subject. Airway segment data were then grouped by lobes for analysis. Each point represents the ratio of mucus-occluded bronchial compartments in 13 patients for each bronchial generation/segment.

**Table 2. The relationship between the airway diameter, mucus score, and spirometry measures.**

| Spirometry value | Total bronchial area Din (mm) | Total bronchial area mucus plug (number) |
|---|---|---|
| FEV1 (% predicted) | NS | −1.1 (−1.9, −0.3) |
| | | $R^2 = 0.42$, $P = 0.0162$ |
| FVC (% predicted) | NS | NS |
| FEV1/FVC | NS | −0.6 (−1.1, −0.2) |
| | | $R^2 = 0.51$, $P = 0.0064$ |

NS, not significant

## Discussion

In this study, we explored the Din and mucus plugs from the segmental bronchi (second generation) to the sixth-generation bronchi during asthma exacerbation and in the stable phase. During asthma exacerbation, the most significant change in the Din was observed at the fifth-generation bronchi, and its mean Din was below 2 mm. Also, more than 40% of the airways were occluded by mucus plugs at the fourth- (sub-subsegmental) and fifth-generation bronchi during asthma exacerbation. Because of the decreased Din and mucus plugs, considerable parts of the airways were severely occluded in the middle zone in acute asthma. These findings may help us to understand the pathophysiology of acute asthma and may provide useful information in quantifying asthma severity.

CT is useful for detecting underlying comorbidities and assessing the severity of airway obstruction in acute asthma. With the analysis of HRCT images, we were able to exclude bronchopneumonia, heart failure, and other diseases that mimic asthma exacerbation.

In quantitative CT imaging, the airway parameters of wall thickness percentage (WT%), wall area percent (WA%), and low attenuation area have been well studied concerning lung function or asthma severity [12–16]. Mucus plugs have only been recently recognized as another CT parameter for the severity of asthma [11]. In this study, we focused on Din and mucus plugs that would directly influence the effectiveness of inhalation therapy during asthma exacerbation. Mucus plugs were more frequently observed than we expected during asthma exacerbation on longitudinal images of the airway. The definition of mucus plugs by CT image was the same as that of the report by Dunican et al. [11]. However, a plugged airway in a cross section and longitudinal section could not necessarily be seen by conventional HRCT images. In this study, precise cross-sectional and longitudinal-sectional images could be obtained by MPR for each bronchial segment. Also, we analyzed the peripheral airway to the sixth-generation bronchi.

In a pathological model of bronchial trees, small airways were reported to compromise the airway generations 8–23 [21]. Since airways with a diameter of < 2 mm are classified as small airways, this study covers small airways. Changes of airway lumen area on HRCT and airway reactivity were first described by Herold et al. [22]. Okazawa et al. [23] identified airway narrowing of intermediate-sized airways produced by inhaled methacholine in asthmatic patients. Shimizu et al. reported that the WA% of the fourth- and fifth-generation bronchi correlated well with airflow limitation in older asthmatics [24], and we previously reported that the airway diameter of the fifth-generation bronchi was most significantly correlated with FEV1 in patients with stable asthma [17]. In the present study, because of the small size of the data, we could not show the relationship between Din and the predicted percent of FEV1 or FEV1/FVC. However, we did show that the total number of mucus plugs correlated with the predicted percent of FEV1 and FEV1/FVC during the stable phase. It supports the previous observation [11] that mucus plugs are associated with airflow limitation in asthma.

In this study, we found that the Din in the fifth-generation bronchi reduced notably during asthma exacerbation. Also, mucus plugs were predominantly observed in sub-subsegmental (fourth-generation) and the fifth-generation bronchi during asthma exacerbation as well as in the stable phase. These observations are partly in accordance with the findings of a previous study that reported that most mucus plugs were in subsegmental bronchi in patients with severe asthma [11]. These data suggest that the fourth- and fifth-generation bronchi are functionally critical areas in bronchial asthma. Further studies on the severity of airway inflammation per airway generation are needed to answer why morphometrical changes are most significant in these levels of bronchi in patients with asthma.

The airway mucus in healthy people is normally a lightly cross-linked gel that does not form plugs [25–26], and why mucus plugs are so common in asthma and why mucus plugs are less frequently observed in more distal airways than in more proximal ones remain to be determined. It is noteworthy that, if there is no surfactant on the airway surface, a small quantity of liquid would move to a narrower part of the airways and eventually block the airway lumen with surface tension [27]. It is reported that the stability of the small airway is maintained by the airway surfactant [28, 29], and the surface activity of the sputum is disrupted in conditions of acute asthma [30, 31]. Therefore, the airway surfactant from the alveolus might partly explain why there are more mucus plugs in the fourth- and fifth-generation bronchi than in sixth-generation bronchi.

There are several limitations to this study that should be noted.

## Accuracy of measurements

It is well accepted that recent CT scanners can correctly measure 1.5-mm airways [32]. In our previous study, differences in phantom tubes (Din, 1.5–4.9 mm) and CT data were 0.1–0.2 mm for Din, and their coefficient of variation (CV) was 1.14%–2.14% [17]. Because of the small differences in repeated measurements, we considered that it was enough to compare the Din in stable phase asthma and asthma exacerbation. However, we avoided using the airway luminal area and WA% to minimize the CV.

## Other limitations

First, the number of study patients was limited. Performing a CT study under the same CT scanner/condition was not possible with a multicenter study, and only part of the asthma exacerbations were studied because of the priority of treatments. To compare the Din at the stable phase and exacerbation, patients had to hold their breath at full inspiration for a while. Therefore, the patients who could not hold their breath at full inspiration could not enter this study. For this reason, the interpretation of the data according to the severity of the asthma attack is limited. Second, the Din during an asthma exacerbation should be much smaller than those presented because we measured the Din 1 hour after the initiation of treatment. Specifically, Din in the fourth- and fifth-generation bronchi should be much smaller because the Din in the compartments with full mucus plugs could not be measured. Because of these limitations, the changes in the Din during an asthma attack should have only been moderately assessed. However, it would be enough to show the relative site of airway obstruction and how severe the airway obstruction is during asthma exacerbation. Finally, our data do not exclude the possible pathological significance in the more distal part of the sixth-generation bronchi.

Our data provide evidence that severe airway obstruction occurs at the fourth- and fifth-generation bronchi in acute asthma. During asthma exacerbation, these airways may become less than 2 mm in diameter or occluded by mucus plugs. Nonetheless, it should be important to note that the first-line treatment for asthma exacerbation is the use of inhalers [1], and they

act directly on obstructed airways. This study shows considerable mucus plugs in acute asthma, but all the patients in this study were successfully relieved by inhaled bronchodilators and systemic corticosteroids. It might be that even partial responses to the inhalers were effective to relieve dyspnea in asthmatics. Mucus plugs can be another target of therapy in acute asthma, as well as in the stable phase.

## Supporting information

**S1 Fig. Ratio of mucus-occluded bronchial compartments in 13 subjects during asthma exacerbation and the stable phase observed from the second- (segmental) to sixth-generation bronchi in the upper, middle/lingula, and lower lobes.** Mucus occlusions were measured by HRCT using curved MPR software. Mucus occlusions were counted per airway generation in all segments of the lung for each subject. Airway segment data were grouped by lobes and by airway generation for analysis. Each point represents the ratio of mucus-occluded compartments in 13 patients for each segment.
(TIF)

**S2 Fig. Ratio of mucus-occluded bronchial compartments in 13 subjects during asthma exacerbation and in the stable phase by lung segments.** Mucus occlusion (mucus plug) was measured by HRCT for each subject at suspended end-inspiratory volume using curved MPR software. Airway segment data per airway generation are expressed. The black bar represents the ratio of mucus occlusion in the stable phase, and the black bar plus gray bar represents the ratio of mucus occlusion during asthma exacerbation.
(TIF)

## Acknowledgments

We wish to thank the Saitama Cardiovascular and Respiratory Center medical staff who cared for the patients.

## Author Contributions

**Conceptualization:** Yotaro Takaku, Kazuyoshi Kurashima.

**Data curation:** Yuki Yoshida, Yotaro Takaku, Yasuo Nakamoto, Noboru Takayanagi, Tsutomu Yanagisawa, Hajime Takizawa, Kazuyoshi Kurashima.

**Formal analysis:** Yuki Yoshida, Yotaro Takaku, Kazuyoshi Kurashima.

**Investigation:** Yuki Yoshida, Yotaro Takaku, Yasuo Nakamoto, Noboru Takayanagi, Tsutomu Yanagisawa, Hajime Takizawa, Kazuyoshi Kurashima.

**Methodology:** Yuki Yoshida, Yotaro Takaku, Kazuyoshi Kurashima.

**Project administration:** Yuki Yoshida, Yotaro Takaku, Kazuyoshi Kurashima.

**Supervision:** Yuki Yoshida, Yotaro Takaku, Kazuyoshi Kurashima.

**Validation:** Yuki Yoshida, Yotaro Takaku, Kazuyoshi Kurashima.

**Visualization:** Yuki Yoshida, Yotaro Takaku, Kazuyoshi Kurashima.

**Writing – original draft:** Yuki Yoshida, Yotaro Takaku, Kazuyoshi Kurashima.

**Writing – review & editing:** Yuki Yoshida, Yotaro Takaku, Yasuo Nakamoto, Noboru Takayanagi, Tsutomu Yanagisawa, Hajime Takizawa, Kazuyoshi Kurashima.

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
