## [Decision Letter · Decision Letter 0]

30 Oct 2019

PONE-D-19-25209

Changes in airway dimensions and mucus plugs in patients with asthma exacerbation

PLOS ONE

Dear Dr Takaku,

Thank you for submitting your manuscript to PLOS ONE. After careful consideration, we feel that it has merit but does not fully meet PLOS ONE’s publication criteria as it currently stands. Therefore, we invite you to submit a revised version of the manuscript that addresses the points raised during the review process.

The Reviewers have pointed to a number of issues that should be amended prior to resubmission.

We would appreciate receiving your revised manuscript by Dec 14 2019 11:59PM. To enhance the reproducibility of your results, we recommend that if applicable you deposit your laboratory protocols in protocols.io, where a protocol can be assigned its own identifier (DOI) such that it can be cited independently in the future. For instructions see: http://journals.plos.org/plosone/s/submission-guidelines#loc-laboratory-protocols

We look forward to receiving your revised manuscript.

Kind regards,

Josué Sznitman

Academic Editor

PLOS ONE

Journal Requirements:

2. We noticed you have some minor occurrence(s) of overlapping text with the following previous publication(s), which needs to be addressed:

https://doi.org/10.1111/j.1440-1843.2011.02052.x

doi: 10.1097/LBR.0000000000000020

In your revision ensure you cite all your sources (including your own works), and quote or rephrase any duplicated text outside the Methods section. Further consideration is dependent on these concerns being addressed.

3. Please include in your Methods section the date ranges over which you recruited participants to this study.

Reviewers' comments:

Reviewer's Responses to Questions

**Comments to the Author**

1. Is the manuscript technically sound, and do the data support the conclusions?

Reviewer #1: Yes

Reviewer #2: Yes

Reviewer #3: Yes

2. Has the statistical analysis been performed appropriately and rigorously? 

Reviewer #1: I Don't Know

Reviewer #2: Yes

Reviewer #3: Yes

3. Have the authors made all data underlying the findings in their manuscript fully available?

Reviewer #1: Yes

Reviewer #2: No

Reviewer #3: No

4. Is the manuscript presented in an intelligible fashion and written in standard English?

Reviewer #1: No

Reviewer #2: No

Reviewer #3: No

5. Review Comments to the Author

Reviewer #1: In this paper, Yoshida et al. investigated the utility of using Chest CT scans to evaluate airway diameter and mucus plugs in a small cohort of patients with asthma both during and after an exacerbation. This paper builds on growing evidence from several groups that CT scans provide useful information in quantifying asthma severity, and can possibly provide a new dimension of asthma heterogeneity (e.g. the CT-derived “mucus score”, recently reported by Dunican et al.).

Yoshida et al. studied 13 asthmatics one hour after initial presentation for an asthma exacerbation, as long as they were able to lie flat and sustain a breath hold during CT imaging. Asthma diameter was quantified using curved multiplanar reconstruction software, and mucus plugs were identified as areas of airway opacification. The main findings were that multiple airways were narrower and more occluded during an exacerbation, as compared to steady state. Although these findings are predictable and not surprising in and of themselves, they add further support to the idea that CT scans can be useful adjuncts to other measures of asthma severity.

The paper is succinctly written with clear figures and results. The manuscript would benefit from careful proofreading and correction for English grammar (starting with the introductory sentence of the Background).

A few additions and modifications would strengthen the impact of the paper.

1) Please provide the length of time between the exacerbation phase and the recovery phase CT scans.

2) Please explain how their quantification of airway occlusion from mucus plugs distinguished between opacified airways, adjacent blood vessels, or possibly airway collapse.

3) How does their method of counting mucus plugs compare or contrast with the “mucus score” reported by Dunican et al (reference 11)?

4) Please provide more information about the subjects studied, including: (i) medication use, (ii) timing of lung function tests relative to CT imaging, (iii) other lung function data (e.g. RV/TLC ratio or bronchodilator reversibilty), and (iv) any data on blood eosinophils or FeNO?

5) It would be more informative to represent the data using individual data points as well as averages, and not just summary bar graphs.

6) Was there any correlation between their CT indices of airway narrowing, or mucus plugs, with lung function measurements, or severity of the asthma exacerbation?

Reviewer #2: Yoshida et al. present a study of airway obstruction measured by two different indices on CT, in patients with asthma at time of exacerbation and at stable follow-up. The first main finding of the paper is that internal diameters (Din) of the 2nd to 6th generation airways are smaller during asthma exacerbation than at follow-up and that these differences are greater in progressively smaller airways up to the 5th generation. The second main finding of the paper is that mucus plugs were more frequent on CT scan up to the 5th generation in patients during asthma exacerbation than at follow-up . The authors conclude that the greatest obstruction at time of exacerbation was seen in the 4-5th generations and that this obstruction was seen in both decreased Din (from smooth muscle bronchoconstriction?, though authors do not state this directly) and increased mucus plugging. There is a lack of published data in the literature comparing patients longitudinally during exacerbation and at stability, which this paper goes some way to address, though in small numbers.

Major comments

By way of background in the abstract, the authors state that “decreased airway diameter and increased mucus plugs … influence the effectiveness of inhaled drugs”. The reviewer is not aware of data supporting this statement. In the introduction, the authors present a good unbiased review of the literature in this area. At the conclusion of the introduction, the authors refer to “the pitfall of inhalation therapy at asthma exacerbation” which needs to be supported by a reference. The conclusion in the abstract accurately reflect and do not overstate the findings of the paper and do not

In the methodology, the authors describe an automated airway tree segmentation using a spanning tree algorithm. I think this needs to be more descriptive. Is this fully automated or is there any manual component to the measurement as there is with many segmenting algorithms? Are the mucus plug measured or counted manually on the segmented CT scans? Counting is alluded to in a figure legend but not in the methods text. How are the same airways/plugs compared longitudinally across scans? How did the authors handle multiple mucus plugs in the same segment? A more detailed description in the on the algorithm for generating the CT measurements and mucus measurements would be welcome in the main text or supplementary data.

In the statistics section there is no reference to adjusting for any confounding variables.

In the results, the authors claim that the decrease in Din and presence of mucus plugs contribute to airway obstruction during an asthma exacerbation but, although pulmonary function was described in the methods, no PFT data is reported in the results. This is likely because it is not standard practice to perform PFT;s during exacerbation, but it would be nice to show a correlation between FEV1 or FEV1/FVC and 4-5th airway obstruction by DIN, by plugs and by a combination of both. In this way, the authors claim that these CT findings are particularly important for airway obstruction would be supported by the data. There is not data in this study to support the hypothesis that the changes in Din and mucus plugs are influencing the pharmacodynamics of inhalation therapy.

In the discussion, the authors report that mucus plugs were less frequently observed in the more distal airways but this is not reflected in their data. The authors discuss the possible role of surfactant in the aetiology of mucus plugging in the small airway but most airways analysed in this study were not considered small airways (<2 mm).

Minor comments:

The text would benefit from review by a native English speaker. In general, it was well written and there were only a few places in the text where minor changes should be made to the language.

Please clarify what is meant by “The exacerbation severity … classified according to GINA guideline”. Authors should clarify whether they mean severity based on symptoms, peak flow, vital signs on presentation etc.

Number of patients recruited should be reported at the beginning of the results section. A consort diagram showing the number of patients screened, excluded, did not consent etc., could be included.

Asthma control is referred to in the results (well- or partially-controlled) but not defined in the methods.

How was bronchiectasis measured? It is referred to in the results but not the methods.

The authors overstate the importance of these finding in suggesting that patients should be made aware that the limited effectiveness of their inhalers during an acute exacerbation and that they should use a systemic corticosteroid instead. This goes agains current guidelines and is not a message we would want to emphasise at this stage. Rewording should be considered.

Reviewer #3: The manuscript titled “Changes in Airway Diameter and Mucus Plugs in Patients with Asthma Exacerbation” by Yoshida et al. is focused on identifying airway diameter changes and presence of mucus plugs immediately following an asthma exacerbation. To do this, the authors followed a small cohort of stable and unstable asthmatics. High resolution CT (HRCT) images were acquired at baseline and one-hour following the exacerbation. In addition, spirometry tests were collected at baseline. While the study is unique, mainly because of the imaging performed immediately following an exacerbation, there are a few points that should be addressed prior to publication:

My main concern lies in the identification of mucus plugs during the exacerbation phase. How are mucus plus distinguished from airway closure due to hyper-constricted airways? Specifically, why would there be more mucus plugs during an exacerbation than during the stable phase?

Why were the PFTs taking two weeks after the HRCT scans were acquired during the stable phase? Should they have been collected at the same time? Were PFTs attempted during or immediately following the exacerbation?

The sentence on line 193 is confusing: “This study clearly showed that small airway shifted toward proximal at asthma exacerbation.” Is this because the small airways are solely defined based on their diameter and the diameters become smaller because of airway constriction? Should classification of which airways are considered small should be done at baseline to allow for direct comparison?

The CT scans were collected at the end of inspiration – which I assume is FRC + TV? Were the lung volumes verified? Specifically, did the authors verify that the images were acquired at the same lung volume during both the baseline and exacerbation scans?

While the manuscript is clear, and it is easy to understand the data collection and interpretation of the results, I believe that the manuscript could benefit from a careful grammar analysis. For example, the first sentence of the abstract should read something like: “Airway obstruction, due to decreased airway diameters and an increased incidence of mucus plugs, are two structural variables that influences the effectiveness of inhaled drugs during an asthma exacerbation.”

6. PLOS authors have the option to publish the peer review history of their article (what does this mean?). If published, this will include your full peer review and any attached files.

Reviewer #1: No

Reviewer #2: No

Reviewer #3: No

---

## [Author Response · Author response to Decision Letter 0]

20 Dec 2019

Response to the reviewers’ comments

Reviewer #1

1) Please provide the length of time between the exacerbation phase and the recovery phase CT scans.

Answer: Materials and Methods section, Study design and participants in revised text. (lines 76-77) 

Thank you for your comment. We added the following:

　 “There was over 6 months between the CT scans in the exacerbation phase and the recovery phase.” 

2) Please explain how their quantification of airway occlusion from mucus plugs distinguished between opacified airways, adjacent blood vessels, or possibly airway collapse.

Answer: Materials and Methods section, CT data acquisition and image analyses in revised text. (lines 118-121)

Thank you for the comment. We added the following sentence in the text.

“Mucus plugs were distinguished from focal opacified airways, such as airway collapse or adjacent blood vessels, identifying no focal decrease of outer airway caliber and recognizing adjacent blood vessels between opened proximal and distal airways from the mucous area.” 

3) How does their method of counting mucus plugs compare or contrast with the “mucus score” reported by Dunican et al. (reference 11)?

Answer: Discussion section in revised text. (lines 286-291)

Thank you for the comment. We added the following sentence in the text.

“The definition of mucus plugs by CT image was the same as that of the report by Dunican et al. [11]. However, a plugged airway in a cross section and longitudinal section could not necessarily be seen by conventional HRCT images. In this study, precise cross-sectional and longitudinal-sectional images could be obtained by MPR for each bronchial segment. Also, we analyzed the peripheral airway to the sixth-generation bronchi.” 

4) Please provide more information about the subjects studied, including (i) medication use, (ii) timing of lung function tests relative to CT imaging, (iii) other lung function data (e.g., RV/TLC ratio or bronchodilator reversibility), and (iv) any data on blood eosinophils or FeNO.

Answer: Table 1 and Materials and Methods section in revised text. (lines 139-140)

Thank you for the comment. The matters that you pointed out about (i) medication use, (iii) other lung function data on bronchodilator reversibility, and (iv) any data on blood eosinophils are now included in Table 1. The matter that you pointed out about (ii) timing of lung function tests relative to CT imaging is now addressed in the revised text as follows: “Pulmonary function tests (PFTs) were performed within two weeks of obtaining the HRCT scans in the stable phase.”

5) It would be more informative to represent the data using individual data points as well as averages, and not just summary bar graphs.

Answer: Figures 3, 4, 5, and S1 as well as Figure Legend in revised text.

Thank you for the comment. We changed the figures by using summary bar graphs and individual data points as well as averages. Each point represents the mean Din ratio of each segment or the percentage of mucus plugs in each segment of 13 subjects.

6) Was there any correlation between their CT indices of airway narrowing, or mucus plugs, with lung function measurements, or severity of the asthma exacerbation?

Answer: Table 2, Result, Discussion section in revised text.

Thank you very much for this important comment.

Lung function and CT indices were analyzed in the stable phase and added to Table 2, the results (second paragraph from the last: lines 240-245), and the discussion section (third paragraph: lines 301-305). The results of the CT indices and severity of asthma exacerbation are presented in the last paragraph of the results section (lines 259-265).

Response to the reviewers’ comments

Reviewer #2

Major comments

By way of background in the abstract, the authors state that “decreased airway diameter and increased mucus plugs … influence the effectiveness of inhaled drugs”. The reviewer is not aware of data supporting this statement. In the introduction, the authors present a good unbiased review of the literature in this area. At the conclusion of the introduction, the authors refer to “the pitfall of inhalation therapy at asthma exacerbation” which needs to be supported by a reference. The conclusion in the abstract accurately reflect and do not overstate the findings of the paper and do not

Answer: Abstract, Introduction section in revised text.

Thank you for this comment.

We changed the background section in the abstract as follows: (line 19)

“Airway obstruction, due to decreased airway diameter and increased incidence of mucus plugs, has not been directly observed at asthma exacerbation.”

We also changed the introduction section as follows: (lines: 54-55)

“Thus, the airway dimeter and mucus plug are two critical factors to limit the effectiveness of inhaled corticosteroids (ICS) and bronchodilators at asthma exacerbations. The mucus plugs, as well as narrowed airways, could be the target of inhaled corticosteroids and bronchodilators during asthma exacerbation.”

In the methodology, the authors describe an automated airway tree segmentation using a spanning tree algorithm. I think this needs to be more descriptive. Is this fully automated or is there any manual component to the measurement as there is with many segmenting algorithms? Are the mucus plug measured or counted manually on the segmented CT scans? Counting is alluded to in a figure legend but not in the methods text. How are the same airways/plugs compared longitudinally across scans? How did the authors handle multiple mucus plugs in the same segment? A more detailed description in the on the algorithm for generating the CT measurements and mucus measurements would be welcome in the main text or supplementary data.

Answer: Materials and methods section, CT data acquisition and image analyses in revised text.

Thank you for the comment.

The trace of bronchial trees was half manually. As described in the text, one　bronchus was selected on the principle that it should be more centered and well recognized in one segment in the stable and acute phase. If the bronchus was obstructed by a mucus plug before the sixth-generation bronchus, we traced the bronchus manually to the sixth-generation bronchus. We recorded the mucus plugs as “yes” or “no” from the second- to the sixth-generation bronchial compartments in a single segment. We then calculated the percentage of mucus plugs among the 13 subjects in a given bronchial generation in a given lung segment.

In the statistics section there is no reference to adjusting for any confounding variables.

　Answer: Materials and methods section, CT data acquisition and image analyses in 　　

revised text. (lines: 122-125)

We added the following comments in materials and methods section to show how we evaluated the Din and mucus plug.

“We noted the presence of mucus plugs as “yes” or “no” from the second- to 

sixth-generation bronchial compartments in one selected airway in one segment. We could then calculate the percentage of mucus plug among the 13 subjects in a given bronchial generation in a given lung segment.” 

In the results, the authors claim that the decrease in Din and presence of mucus plugs contribute to airway obstruction during an asthma exacerbation but, although pulmonary function was described in the methods, no PFT data is reported in the results. This is likely because it is not standard practice to perform PFT;s during exacerbation, but it would be nice to show a correlation between FEV1 or FEV1/FVC and 4-5th airway obstruction by DIN, by plugs and by a combination of both. In this way, the authors claim that these CT findings are particularly important for airway obstruction would be supported by the data. There is not data in this study to support the hypothesis that the changes in Din and mucus plugs are influencing the pharmacodynamics of inhalation therapy.

　Answer: Results and Discussion section in revised text.

Thank you for the comment.

We did not perform a PFT in patients with acute asthma. We performed PFT in patients in the stable phase, and the data were used only for patient characteristics, Table 1. It is well known that Din correlates with PFT data, but we did not analyze the correlation in this study because the size of the study was too small.

In the revised manuscript, we analyzed the relationship between PFT data and CT image data. To our surprise, the mean Din of all bronchial compartments nor the mean Din of the second to the sixth airway was not correlated with the predicted percentage FEV1 or FEV1/FVC, however, the total number of mucus-plug-positive bronchial compartments were correlated with the predicted percentage FEV1 and FEV1/FVC (Table 2 was added). The data further supports the results of Dunican’s paper (reference 11).

In the discussion, the authors report that mucus plugs were less frequently observed in the more distal airways but this is not reflected in their data. The authors discuss the possible role of surfactant in the aetiology of mucus plugging in the small airway but most airways analysed in this study were not considered small airways (<2 mm).

Answer: Results and Discussion section in revised text. (line: 323-325)

Thank you for the comment.

We discussed the surfactant to explain the results of Figs. 4 and S2. We changed the section in the discussion as follows:

 　“Therefore, the airway surfactant from the alveolus might partly explain why there are more mucus plugs in the fourth- and fifth-generation bronchi than in sixth-generation bronchi.”

Minor comments:

The text would benefit from review by a native English speaker. In general, it was well written and there were only a few places in the text where minor changes should be made to the language.

Answer:

Thank you for the comment.

The revised text was reviewed by a native English speaker.

Please clarify what is meant by “The exacerbation severity … classified according to GINA guideline”. Authors should clarify whether they mean severity based on symptoms, peak flow, vital signs on presentation etc.

Answer: Materials and methods section, Study design and participants in revised text.

(lines: 81-86)

Thank you for the comment. We added the text as follows:

“The severity of asthma exacerbation in this study’s patients was determined according to the descriptions in the section “Management of asthma exacerbations in the emergency department” in the GINA guidelines. We used objective assessment and other measurements, such as respiratory rate, pulse rate, O2 saturation, and accessory muscles being used, for the determination.”

Number of patients recruited should be reported at the beginning of the results section. A consort diagram showing the number of patients screened, excluded, did not consent etc., could be included.

Answer: Result section in revised text. (lines: 161-163)

Thank you for the comment. We added the following sentence to the result section:

“Seventeen patients who had emergency visits to our center with asthma attacks were included. Of these, three were excluded because of their life-threatening condition. One did not consent, and the remaining 13 patients were enrolled.”

Asthma control is referred to in the results (well- or partially-controlled) but not defined in the methods.

Answer: Materials and methods section, Study design and participants in revised text.

(lines: 78-79)

Thank you for the comment. We added the following sentence in the methods section,

“We evaluated asthma control according to the consensus-based GINA symptom control tool.”

How was bronchiectasis measured? It is referred to in the results but not the methods.

Answer: Materials and methods section, CT data acquisition and image analyses in revised text. (lines: 121-122)

Thank you for the comment.

We added the following sentence in the materials and methods section, CT data acquisition and image analysis.

“We checked whether mucus plugs were associated with bronchiectasis, defined as a bronchoarterial ratio of greater than 1.5.” 

The authors overstate the importance of these finding in suggesting that patients should be made aware that the limited effectiveness of their inhalers during an acute exacerbation and that they should use a systemic corticosteroid instead. This goes agains current guidelines and is not a message we would want to emphasise at this stage. Rewording should be considered.

Answer: Discussion section in revised text.

Thank you very much for the important comment.

“Nonetheless, it should be important to note that the first-line treatment for asthma exacerbation is the use of inhalers [1], and they act directly on obstructed airways. This study shows considerable mucus plugs in acute asthma, but all the patients in this study were successfully relieved by inhaled bronchodilators and systemic corticosteroids. It might be that even partial responses to the inhalers were effective to relieve dyspnea in asthmatics.”

We changed the first paragraph of the introduction and added the above comment to the conclusion part of the discussion (lines: 351-357).

Response to the reviewers’ comments

Reviewer #3

My main concern lies in the identification of mucus plugs during the exacerbation phase. How are mucus plus distinguished from airway closure due to hyper-constricted airways? Specifically, why would there be more mucus plugs during an exacerbation than during the stable phase?

Answer: Materials and methods section, CT data acquisition and image analyses in revised text. 

Thank you for the comment.

We added the following sentence to distinguish mucus plugs from airway closure.

“Mucus plugs were distinguished from focal opacified airways, such as airway collapse or adjacent blood vessels, identifying no focal decrease of outer airway caliber and recognizing adjacent blood vessels between opened proximal and distal airways from the mucous area.” (lines: 118-121)

Increased mucus plugs were reported in fatal asthma [9, 10], and its mechanisms are considered as type 2 inflammation, as noted in the discussion [11].

Why were the PFTs taking two weeks after the HRCT scans were acquired during the stable phase? Should they have been collected at the same time? Were PFTs attempted during or immediately following the exacerbation?

Answer:

Thank you very much for the comment.

It is desirable to test CT and PFTs on the same day, and they were done on the same day in five patients. However, we also considered it acceptable if they were done within two weeks.

The sentence on line 193 is confusing: “This study clearly showed that small airway shifted toward proximal at asthma exacerbation.” Is this because the small airways are solely defined based on their diameter and the diameters become smaller because of airway constriction? Should classification of which airways are considered small should be done at baseline to allow for direct comparison?

Answer: Discussion section, third paragraph in revised text.

Thank you for the comment.

To our knowledge, the original definition of “small airways” is solely defined based on their diameter. However, what we would like to say is that sixth-generation bronchi are less than 2 mm in diameter in the stable phase and during exacerbation. We changed the section as follows:

“In a pathological model of bronchial trees, small airways were reported to comprise 

the airway generations 8–23 [21]. Since airways with a diameter of < 2 mm are classified as small airways, this study covers small airways”. the zone of small airways may shift according to the changes of airway diameter that occur in asthma. This study clearly showed that small airway shifted toward proximal at asthma exacerbation.” 

The CT scans were collected at the end of inspiration – which I assume is FRC + TV? Were the lung volumes verified? Specifically, did the authors verify that the images were acquired at the same lung volume during both the baseline and exacerbation scans?

　Answer: Materials and methods section, CT data acquisition and image analyses in 　　

revised text.

Thank you for the comment.

The CT images were acquired at the same lung volume during the baseline and exacerbation at the end of inspiration. We checked the lung volume by the lung size and lung architecture in the same lung slice in a CT image.

We changed the sentence as follows: (lines 99-101)

“A 256-slice CT scanner (Brilliance iCT; Philips Healthcare, Cleveland, OH, USA) was used as previously described [17]. The CT scans were obtained from the suspended end-inspiratory volume at baseline and during exacerbation. We also checked lung size and lung architecture in the same slice to confirm they were obtained at the end-inspiratory volume.” 

While the manuscript is clear, and it is easy to understand the data collection and interpretation of the results, I believe that the manuscript could benefit from a careful grammar analysis. For example, the first sentence of the abstract should read something like: “Airway obstruction, due to decreased airway diameters and an increased incidence of mucus plugs, are two structural variables that influences the effectiveness of inhaled drugs during an asthma exacerbation.”

Answer:

Thank you very for the comment.

The revised text was reviewed by a native English speaker.

---

## [Decision Letter · Decision Letter 1]

3 Feb 2020

Changes in airway diameter and mucus plugs in patients with asthma exacerbation

PONE-D-19-25209R1

Dear Dr. Takaku,

We are pleased to inform you that your manuscript has been judged scientifically suitable for publication and will be formally accepted for publication once it complies with all outstanding technical requirements.

With kind regards,

Josué Sznitman

Academic Editor

PLOS ONE

Additional Editor Comments (optional):

Reviewers' comments:

Reviewer's Responses to Questions

**Comments to the Author**

1. If the authors have adequately addressed your comments raised in a previous round of review and you feel that this manuscript is now acceptable for publication, you may indicate that here to bypass the “Comments to the Author” section, enter your conflict of interest statement in the “Confidential to Editor” section, and submit your "Accept" recommendation.

Reviewer #1: All comments have been addressed

Reviewer #3: All comments have been addressed

2. Is the manuscript technically sound, and do the data support the conclusions?

Reviewer #1: Yes

Reviewer #3: Yes

3. Has the statistical analysis been performed appropriately and rigorously? 

Reviewer #1: Yes

Reviewer #3: Yes

4. Have the authors made all data underlying the findings in their manuscript fully available?

Reviewer #1: Yes

Reviewer #3: No

5. Is the manuscript presented in an intelligible fashion and written in standard English?

Reviewer #1: Yes

Reviewer #3: Yes

6. Review Comments to the Author

Reviewer #1: The authors adequately addressed all of the comments raised during the initial review. The inclusion of individual data points and new data about patients strengthens the impact of the manuscript.

Reviewer #3: (No Response)

7. PLOS authors have the option to publish the peer review history of their article (what does this mean?). If published, this will include your full peer review and any attached files.

Reviewer #1: No

Reviewer #3: Yes: Jessica M Oakes

---

## [Editor Report · Acceptance letter]

13 Feb 2020

PONE-D-19-25209R1 

Changes in airway diameter and mucus plugs in patients with asthma exacerbation 

Dear Dr. Takaku:

I am pleased to inform you that your manuscript has been deemed suitable for publication in PLOS ONE. Congratulations! Your manuscript is now with our production department. 

With kind regards,

on behalf of

Prof. Josué Sznitman 

Academic Editor

PLOS ONE